# Evaluation of Laboratory Methods of Determination of SBS Content in Polymer-Modified Bitumens

**DOI:** 10.3390/ma13225237

**Published:** 2020-11-19

**Authors:** Maria Ratajczak, Artur Wilmański

**Affiliations:** 1Institute of Building Engineering, Faculty of Civil and Transport Engineering, Poznan University of Technology, Piotrowo St. No. 5, PL-60965 Poznan, Poland; 2Road Laboratory, Technology Department, General Directorate for National Roads and Motorways, Hawelanska St. No. 12, PL-60763 Poznan, Poland; awilmanski@gddkia.gov.pl

**Keywords:** IR spectroscopy, quantification, polymer-modified bitumen, SBS content

## Abstract

The article deals with the issue of determination of the content of SBS (Styrene-Butadiene-Styrene) in polymer-modified bitumens (PMBs). The effect of SBS copolymer on the physical and rheological properties of bitumens has been thoroughly investigated and widely described in the literature. Condition surveys of structures and evaluation of the properties of materials used at construction sites have become a huge challenge for construction engineering. Determination of the content of SBS modifier in various building materials (asphalt mixtures and bituminous waterproofing compounds) is a good example in this respect. Based on the laboratory tests, mid-infrared spectroscopy was found to be the most effective analytical method. It can be used for easy detection of the presence of SBS in a modified bitumen. However, quantitative analysis is an issue that calls for research. Currently, there are no standard guidelines, whether national or European, that would regulate the method of testing. Three test methods were assessed in this study: the AASHTO T302–15 standard method and two Australian methods described in codes of practice (T521 and Q350) developed by the local authorities, which define a standard way of determining the amount of SBS in polymer-modified bitumens. The tests were carried out on standard controls and samples sourced from the industry. The above-mentioned test methods were assessed in terms of accuracy of determination, reliability of results obtained on the industrial samples, level of complexity of the test procedure, sample preparation techniques and the type of the required reagents.

## 1. Introduction

Bitumen became a popular building material quite recently, namely at the beginning of the 20th century and already at that time the first modification attempts to improve its performance were made [1,2,3]. Systematic research on the effect of various admixtures on the properties of bitumens began in the late 1960s and early 1970s. The factors that gave momentum to this research were the 1973 oil crisis and rapid development of polymer chemistry in the late 1970s and early 1980s [4,5,6]. For almost thirty years, styrene-butadiene-styrene (SBS) has been the most commonly used bitumen modifier. The purpose of modification is to improve the bitumen’s properties, widen the viscoelastic region of the bitumen behavior or increase its resistance to permanent deformation [5,7,8]. The history of SBS as a modifier started with the need to recycle waste, namely scrap tires, which after grinding into powder were added to bitumen. The first versions of SBS, produced as an alternative to rubber powder, were tested as early as the 1960s. At first, SBS was used primarily for production of sports footwear. It was no sooner than in the mid-1990s that SBS obtained the form and achieved the performance of the material we know today [9,10]. SBS is produced in the form of a fine powder, balls, crumbs, bales or a latex liquid [11], depending on the method of polymerization used in the process. Each bitumen has a slightly different group composition and therefore the effects of modification with SBS are slightly different in each case [3,4,12]. The application of SBS as a bitumen modifier was the subject of research of many studies carried out by numerous scientific and industrial research centers, and, as a result, its effect on the physical and rheological properties of bitumens is well known and broadly described in the literature [13,14,15,16,17,18,19,20,21].

The present state of engineering knowledge makes it possible to precisely design a civil structure or manufacture a material with the desired properties. Currently, research efforts are focused on the issues related to condition assessment of structures and evaluation of the properties of construction materials available on the market. To build or make something is no longer a problem for an engineer. Conversely, to find out what that something was made of is still a challenge. In this respect, our research concerned the determination of the percentage content of SBS in polymer-modified bitumens used to produce a specific HMA and waterproofing compound. As follows from various laboratory studies [22,23,24,25,26,27,28,29,30,31], mid-infrared spectroscopy appears to be the most effective method for experimental determination of the SBS content in bitumens. The first attempts of this kind were undertaken in the early 1990s and described by Choquet and Ista [22]. Although this issue has been investigated at different research centers throughout the world [23,24,25,26,27,28], in most cases, the results boil down to the conclusion that spectroscopic methods could be successfully used to determine the content of SBS in polymer-modified bitumens. However, no clear and unequivocal answer on how to do it was provided. In 1993, the first such test procedure was developed by the Roads and Maritime Services, a state government agency of New South Wales, Australia, intended to be used locally and for internal purposes only. The most recent revision of this protocol [32], issued in 2012, is still a local standard test procedure applied in the area of News South Wales, Australia. A similar document can be found among the internal guidelines of the Department of Transport and Main Roads of the Queensland government, Australia [33], most recently published in 2014. In 2005, AASHTO published its standard test procedure No. T302-05, providing a detailed description of the test procedures for determination of SBS content in polymer-modified bitumens through IR spectroscopy. The most recent issue was published in 2019 under a different number: AASHTO T302–15 (2019) Standard Method of Test for Polymer Content of Polymer-modified Emulsified Asphalt Residue and Asphalt Binder [34].

Neither the European Standards nor national codes provide a standard test method for determining the content of SBS in polymer-modified bitumens. In this study, we assessed the effectiveness of the above-mentioned methods through multi-criteria evaluation. The primary evaluation criterion was the accuracy of measurement and the reliability of the results obtained for the industrial samples. The complexity of the test procedure, sample preparation technique and the type of reagents needed to carry out the test were also analyzed.

## 2. Materials and Methods

### 2.1. Materials

The laboratory testing part of this research was carried out in two stages. In the first stage, in-house samples were used, prepared by modification of base bitumens with different amounts of SBS. In the second stage, the developed methods of determination of SBS content in modified bitumens were verified through testing of industrial samples. The samples were obtained from commercial asphalt mixing plants, which used polymer-modified bitumens for the production of asphalt mixtures. In the first stage, the base binders were paving bitumens 160/220 and 50/70. These bitumens were produced by PKN ORLEN S.A. (Warsaw, Poland) from crude oil coming from the Ural oil fields. They are widely used in the production of waterproofing and paving materials. The main properties of these basic bitumens are compiled in Table 1, and their group compositions are given in Figure 1 together with the calculated values of the colloidal stability index GI. As compared to bitumen 160/220, bitumen 50/70 featured a considerably lower penetration value, higher softening point and higher breaking point. Bitumen 50/70, evidently harder and more brittle, contained less saturates and more asphaltenes than bitumen 160/220.

This was reflected in the values of the colloidal stability index (GI), which were 0.31 in the case of bitumen 160/220 and 0.48 in the case of bitumen 50/70. Based on the literature review, the value of GI = 0.36 is considered the limit value [4,8]. Bitumens of GI < 0.36 feature good compatibility with polymer-based modifiers. Selection of bitumens with different values of the colloidal stability index (compatible or incompatible with the polymer) was another aspect considered in the determination of the content of SBS.

The base bitumens were modified with KRATON^TM^ D1101 A Polymer, an SBS with a linear structure, manufactured by Kraton Corporation (Wesseling, Germany). The main properties of SBS used in this study are given in Table 2. KRATON^TM^ D1101 is one of the most widely used polymers in the Polish building materials market. The test results presented in this paper are the first part of research project. In the next stage, different types of SBS and bitumen from different sources will be tested to evaluate the considered test methods.

Bitumens 160/220 and 50/70 were modified by addition of 1%, 1.5%, 2%, 2.5%, 3%, 3.5%, 4%, 6%, 8% and 10% of SBS by weight. The method of preparation of lab samples is depicted in the schematic in Figure 2. IKA T 25 Digital disperser (IKA^®^-Werke GmbH & CO. KG, Staufen, Germany), equipped with a special dispersing tool, was used. The instrument offers speed range up to 25,000 rpm and can handle high viscosity materials of up to 5 Pa·s.

In the second stage of the laboratory tests, five types of polymer-modified bitumens used in HMA production were tested: three classified as polymer-modified bitumens (PMB) and two belonging to the highly-modified asphalt (HiMA) group. For each bitumen type, the tests were done on samples of fresh binder, binder subjected to simulated short-term ageing according to EN 12607-1 (RTFOT) [36] and binder recovered from HMA produced with the use of the bitumen in question. The binder was recovered according to the procedure described in EN 12697-3 [37] using toluene as solvent. The effect of ageing of bitumen can affect the bitumen spectra, and it can be evaluated through IR spectroscopy [38,39,40]. To take this into consideration, fresh bitumen directly from manufacturer, binder subjected to simulated short-term ageing (RTFOT) and binder recover from HMA were tested. Table 3 gives a list of samples used in the tests and their symbols, by which they are referred to further in this article.

### 2.2. T302–15 (2019) Standard Method of Test for Polymer Content of Polymer-Modified Emulsified Asphalt Residue and Asphalt Binders 

Three test methods were applied for determining the content of SBS in polymer-modified bitumens, which were then assessed by the application of the criteria including accuracy of measurement, complexity of the test procedure, sample preparation technique and harmfulness of the reagents.

The first of the assessed methods is described in the code of practice No. T302–15 (2019)—Standard Method of Test for Polymer Content of Polymer-modified Emulsified Asphalt Residue and Asphalt Binder [34] developed by the American Association of State Highway and Transportation Officials (AASHTO). This test is used for determining the percent concentration of SBR (styrene-butadiene rubber), SB (styrene-butadiene) or SBS polymers in polymer-modified bitumens. This code offers two methods of measurement of the infrared spectrum: attenuated total reflectance (ATR) and measurement in the transmission mode (T). In the first sampling technique, the samples were prepared by heating the binder to its flow temperature, however not exceeding 163 °C. After transition to the liquid state, the sample was stirred to obtain a uniform, homogenous mixture and then spread on paper to obtain a ca. 1 mm thick layer, sufficient to cover entirely the face of the ATR crystal. The sample was left to cool and then placed on and pressed against the diamond crystal to push out any air bubbles from the sample and ensure complete contact between the bitumen layer and the face of the crystal. For measurement in the transmission mode, the sample of bitumen was stirred and 1 g of the sample was placed in a glass vial and left to cool down to room temperature. It was then dissolved in ca. 10 mL of a solvent (trichloroethane or tetrahydrofuran can be used for this purpose). Next, the solution was applied on the appropriate medium using a dropper. After the application, the sample was left at room temperature until complete evaporation of the solvent. In this project, tetrahydrofuran (THF) was used as solvent and KBr discs were used as the medium. To check the effect of the film thickness on the results, different quantities of the material were applied on the discs, namely one drop (Specimens 1d), three drops (Specimens 3d) and five drops (Specimens 5d). The percentage of SBS was determined by calculating the peak height or area ratio between the polymer and bitumen peaks. The peak at 965 cm^−1^ wave number (A1) was considered indicative of the presence of SBR, SB or SBR with the base bitumen peak at the wave number of 1375 cm^−1^ (A2). Figure 3 represents example PMB infrared scans and the method of determination of the values of A1 and A2. Following determination of the values of A1 and A2, the percentage concentration of polymer was determined using the calibration curve and given with 0.1% accuracy (Figure 4).

### 2.3. T521 Quantification of Polymer-Modified Binders Using Infrared Spectrum

The second assessed test method is described in the code of practice No. T521 Quantification of polymer-modified binders using infrared spectrum [32], issued by the Roads and Maritime Services, a state government agency of New South Wales, Australia. This test method enables determination of the percentage content of the most common polymers used for modification of bitumens, namely SBS and SBR or EVA and EMA. Below are the main assumptions adopted in this method of quantification:The C/H ratio of bitumen is constant.The band of the CH_2_ functional group is taken as a standard.Bands 1 and 2 at wave numbers 700 and 970 cm^−1^, respectively, are indicative of the presence of SBS and SBR modifiers, yet without discriminating them.In principle, a band at 700 cm^−1^ (1) can be indicative of the presence of any mono substituted aromatic compound. The simultaneous presence of Bands 1 and 2 (700 and 970 cm^−1^) is very strong evidence of the presence of styrene-butadiene copolymer combined at a 30:70 ratio (note that lab samples were modified with SBS containing 30–32% styrene).

The samples were prepared by heating them to 175 ± 5 °C according to EN 12594 [41]. After stirring the mix, ca. 1 g of bitumen was transferred to a beaker and allowed to cool down. Next, ca. 5 mL of toluene were added, and the mix was stirred until completely dissolved. Next, some amount of the solution, “a drop or two”, was applied on the disc by means of a dropper and the disc was left to rest to allow the solvent to evaporate. When the applied sample formed an impermeable film, the disc was placed in the laboratory oven heated to 100 ± 5 °C for ca. 10 min and after that transferred to the desiccator to cool down to room temperature. Then, the sample, prepared as described above, was used to measure transmission in the mid infrared band in the transmission mode of measurement. After the infrared spectrum was scanned, the waveband intensity was checked for the wave number 1460 cm^−1^. It should fall in the range of 5–30%. A transmission rate higher than 30% indicates that the residual film is too thin, and another drop must be applied on the disc. If the transmission rate is below 5%, this means that the residual film is too thick and a new sample must be prepared. Determination of the concentration of SBS or SBR in PMBs is based on analyzing three characteristic peaks for wave numbers 700 (1), 970 (2) and 1380 cm^−1^ (3), respectively. To each of these peaks, tangential base lines are drawn for the following bands:690–790 cm^−1^—for peak at 700 cm^−1^930–1130 cm^−1^—for peak at 970 cm^−1^1230–1400 cm^−1^—for peak at 1380 cm^−1^

Next, the percent transmission was read for each peak (*Peak_n_*) and for the plotted baseline (*Line_n_*) at the same wave number with an accuracy to 0.5% and the value of *A_n_* was calculated using the following formula:*A_n_ = log*_10_*(Line_n_) − log*_10_*(Peak_n_)*(1)
where *A_n_* is the result for a given band *n*; *Line_n_* is the baseline transmission value %; *Peak_n_* is the bsolute transmission value in percent; and *n* is the band number, with 1 = band 690–790 cm^−1^, 2 = band 930–1130 cm^−1^ and 3 = band 1230–1400 cm^−1^. Figure 5 represents example PMB infrared scans and the method of determination of the value of *A_n_*.

The percentage content of Styrene was calculated as follows:if *A*_1_*/A*_3_ < 0.57 then *%Styrene = (5.25∙A*_1_)*/A*_3_,(2)
if *A*_1_*/A*_3_*≥ 0.57* then *%Styrene* = 0.3∙{[(12.8∙A_1_)/A_3_] + 2.7}, (3)

The percentage content of Butadiene was calculated with the following formula:*% Butadiene = (10.5A*_2_)*/A*_3_,(4)

The percentage content of SBS or SBR was calculated with the following formula:*%SBS* or *%SBR = % Styrene + % Butadiene*(5)

### 2.4. Q350: SBS Content of Polymer-Modified Binder 

The third assessed test method is described in the code of practice No. Q350: SBS content of polymer-modified binder [33] developed by the Department of Transport and Main Roads of the government of Queensland, Australia. This method employs Attenuated Total Reflection-Fourier Transform Infrared (ATR-FTIR) Spectroscopy for determining the SBS content in PMBs. It can be applied for determining the copolymer content in both PMBs and in hot mix asphalts (HMAs) containing PMBs. The sample preparation process started by transferring ca. 2 g of bitumen to a glass vial, followed by adding to it 10 mL of carbon disulfide (CS_2_). The vial was then placed in a mechanical shaker and shaken for at least 1 h at room temperature. Using a dropper, two drops of the solution taken from the upper part of the vial were transferred onto a glass plate in a fume hood, minimizing the diameter of the resulting spot as far as practicable. Next, the glass plate was left to allow the solvent to evaporate. This sequence was repeated until adequate thickness of the film was obtained (ca. 5 mm diameter of sample and thickness sufficient to block the passage of light). The sample was then placed for 1 h in a drying oven heated to 40 °C. The measurement of mid-infrared spectrum was carried out through the ART method with the test setup parameters given in Table 4.

The percentage of SBS was calculated using the following formula:S = (28.42·P_697_)/(P_966_ + P_911_ + P_80 8_ + P_697_)(6)
where: S is the SBS percentage and P_n_ is the peak area for the wave number *n*.

## 3. Results

### 3.1. T302–15 (2019) Standard Method of Test for Polymer Content of Polymer-Modified Emulsified Asphalt Residue and Asphalt Binders

Figure 6 and Figure 7 give the SBS percentages in bitumens 160/220 and 50/70, determined according to the test procedure described in AASHTO T302–15 using the transmission mode of measurement. The test was performed for samples with tree different dry film thicknesses obtained by applying one drop (1d), three drops (3d) and five drops (5d) of the solution on KBr discs. In the processing of the results, the automatic baseline correction functions were used. Regression analysis was employed to represent, using the most probable mathematical function, the relationship between the actual content of the bitumen modifier and the value obtained experimentally according to the above-described codes of practice. The parameters of the regression model were chosen with the least squares method. The prediction accuracy of the regression function was evaluated by means of the coefficient of determination R^2^. The results obtained for the industrial samples are given in Figure 8.

Figure 9 and Figure 10 give the SBS contents in bitumens 160/220 and 50/70, determined according to the test procedure described in AASHTO T302–15, using the ATR technique. In the processing of the results, the automatic baseline correction functions were used. Due to large differences between the determined and actual concentrations of the modifier, new calibration curve equations were derived on the basis of the A1/A2 ratio. The results obtained for the industrial samples are given in t Figure 11. The SBS content was calculated according to the AASHTO standard curve and the calibration curves derived from the experimental results obtained on the laboratory samples.

### 3.2. T521 Quantification of Polymer-Modified Binders Using Infrared Spectrum

Figure 12 and Figure 13 give the SBS contents in bitumens 160/220 and 50/70, obtained according to the procedure described in the T521 code of practice. The test was performed for samples with three dry film thicknesses, obtained by applying one drop (Specimens 1d), three drops (Specimens 3d) and five drops (Specimens 5d) of the solution on the KBr discs. The results obtained on the industrial samples are given in Figure 14.

### 3.3. Q350: SBS Content of Polymer-Modified Binder

Figure 15 and Figure 16 give the SBS contents in bitumens 160/220 and 50/70 measured on the industrial samples, according to the procedure described in Q350.

## 4. Discussion

### 4.1. T302–15 (2019) Standard Method of Test for Polymer Content of Polymer-Modified Emulsified Asphalt Residue and Asphalt Binders

The SBS contents determined on laboratory samples (Figure 6 and Figure 7) by measurement in transmission mode showed that film thickness had a negligible effect on the obtained result. The mean measurement error for the samples of bitumen 160/220 was in the range of 0.35–0.54% and for samples of bitumen 50/70 in the range of 0.23–0.27%. In all analyzed cases, R^2^ was close to 1. A small variability was noted in the SBS contents measured on the industrial samples (Figure 8) among the controls, samples after RTFOT and samples recovered from HMA. The SBS content was in the range of 2.5–3.8% for PMB and 2.6–3.6% for HiMA. A small difference between values obtained on PMB and HiMA samples may suggest that in the production of HiMA a hybrid modifier with an SBS copolymer content was used.

A considerable spread was noted between the data obtained with ATR and calculated from the nominal standard curve (Figure 9 and Figure 10). The mean measurement errors for bitumens 160/220 and 50/70 were 1.65% and 1.57%, respectively. Smaller variabilities, yet still above 1%, viz. 1.19% and 1.08%, respectively, were obtained with the new standard curves. The values of coefficient of determination R^2^ were also smaller: 0.78 for bitumen 160/220 and 0.81–0.83 for bitumen 50/70. The measurement data obtained on industrial samples (Figure 11) were characterized by greater spread of values as compared to the measurements in transmission mode. The SBS copolymer content was in the range of 2.0–8.2% for PMB and 1.8–4.9% for HiMA. Similar to the transmission mode, in this case, the SBS copolymer contents did not vary between PMB and HiMA.

### 4.2. T521 Quantification of Polymer-Modified Binders Using Infrared Spectrum

The dataset of SBS contents obtained according to the T521 code of practice shows that, for bitumen 160/220 (Figure 12), the required film thickness was obtained for Specimens 1d. For this amount of applied solution, the values of %T for the band of wave number 1460 cm^−1^ were in the range of 5–30%. For bitumen 160/220, the mean measurement error was 0.48% for Specimens 1d and 1.11% and 0.72% for Specimens 3d and 5d, respectively. In addition, for Specimens 1d, the highest value of R^2^ was obtained. In the case of bitumen 50/70 (Figure 13), no significant effect of the film thickness on the accuracy of measurement data was observed, even though the values of %T of below 5% were obtained for Specimens 3d and 5d for the band with wave number 1460 cm^−1^. The mean measurement error was 0.59% for Specimens 1d and 0.26% and 0.24% for Specimens 3d and 5d, respectively. The values of R^2^ were in all cases above 0.9.

In addition, the industrial samples (Figure 14) did not reveal a significant effect of the film thickness on the measurement data, even though values of %T below 5% were obtained for Specimens 3d and 5d for the band with wave number 1460 cm^−1^. No significant differences were found in the results obtained on the samples of fresh binder, binder subjected to RTFOT (rolling thin film oven test) or binder recovered from HMA. The mean SBS contents were 3.6%, 3.8% and 3.1% for bitumens A45/80-55, A45/80-65 and B45/80-55, respectively. In the case of HiMA bitumens, the mean SBS contents were 4.5% and 5.2% for bitumens C45/80-80 and D45/80-80, respectively. In contrast to the results obtained with the procedure described in T302–15, here a difference between PMB and HiMA binders is evident.

### 4.3. Q350: SBS Content of Polymer-Modified Binder

A considerable variability was noted for the values of SBS content obtained on lab samples according to Q350 (Figure 15). The mean measurement errors were 1.41% and 1.79% for bitumens 160/220 and 50/70, respectively. The values of R^2^ were 0.75 for bitumen 160/220 and 0.54 for bitumen 50/70. In the case of industrial samples of PMBs (Figure 16), there was no significant difference between the results obtained on the samples of fresh binder, binder subjected to RTOF or binder recovered from HMA. The mean SBS contents were 5.1%, 4.9% and 4.2% for bitumens A45/80-55, A45/80-65 and B45/80-55, respectively. In the case of HiMA a considerable difference was observed between the results obtained on the samples of fresh binder, binder subjected to RTOF and binder recovered from HMA and the highest values of 7.7 and 7.3 were obtained for bitumens C45/80-80 and D45/80-80, respectively. Similar to the results of measurement performed according to the procedure described in T521, there is a marked difference between the PMB and HiMA results.

### 4.4. Evaluation of the Test Methods

The measurement results were used to compare the test methods listed in Table 5. The accuracy of the results obtained on the laboratory samples and the reliability of the results obtained on the industrial samples were checked first. The complexity of the test procedure, sample preparation technique and type of solvent used in the tests were also taken into consideration.

In the case of the laboratory samples, the highest accuracy of results was obtained with the procedure described in AASHTO T302–15 (transmission method). This method utilizes the height ratio of the polymer and binder peaks and is characterized by simple measurement and data analysis techniques. The most time-consuming part of the process is preparation of the test specimens, comprising preparation of bitumen solution and the discs on which it will be applied. However, simplification of this method by use of ATR (spectral analysis of solid materials) and analysis of peak areas instead of peak heights results in considerable measurement errors in determination of the actual SBS copolymer content. Moreover, the results obtained on the industrial samples raise doubts due to the lack of a clear difference in the content of the modifier between PMB and HiMA binders.

The biggest measurement error was observed in the case of the results obtained according to the procedure described in the Q350 code of practice. In addition, it involves the most time-consuming preparation of samples and the use of carbon disulfide as a solvent, which is a toxic chemical compound, harmful to the central nervous system and has potential for carcinogenic and mutagenic effects. Large measurement errors are caused by the utilization of ATR and determination of the SBS percentages on the basis of peak areas. This is the only method based on the use of as many as four different bands, which might imply measurement errors. However, marked differences in the experimental determinations of SBS percentages in PMB and HiMA are worth noting in this case.

In the T521 method, the mean error of measurement obtained on the lab samples was 0.66%, which was slightly higher than the mean error obtained in the method T described in AASHTO T302–15. Transmission mode of measurement and peak heights are used also in the T521 method and the difference is in the preparation of samples, which involves the use of a different solvent (THF or Toluene). Although an additional band of 690 cm^−1^ wave number (indicative of the presence of SBS) introduced in the spectral analysis in this method increased the mean measurement error in the case of laboratory samples, the considerable difference between the results obtained for PMB and HiMA binders, testifying to its performance and efficiency, can support the choice of this procedure.

## 5. Conclusions

Laboratory determination of the content of SBS copolymer in PMBs is usually performed on samples obtained from the industry with the purpose of assessing the quality of the applied construction materials. It is quite rare that in-house controls can be prepared using fresh binder and its own standard curve be derived. The T302–15 test method with transmission mode yields very satisfactory test results for lab samples, but with industrial samples no differences between PMB and HiMA were observed, which is questionable. The Q350 and T302–15 ATR test methods are characterized by a high mean error value. The T521 test method has satisfactory value of the mean error and the test results for industrial samples are reliable Therefore, in view of the above, the measurement technique described in the T521 code of practice is considered to be the most optimal choice among the methods analyzed in this study.

## Figures and Tables

**Figure 1 materials-13-05237-f001:**
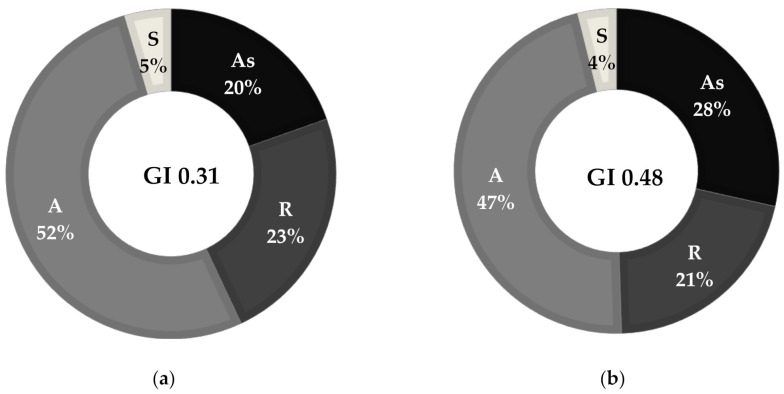
Chemical composition (SARA fractions: S—saturates, A—aromatics, R—resins, As—asphaltenes) and Gaestel Index of bitumens: 160/220 (**a**); and 50/70 (**b**).

**Figure 2 materials-13-05237-f002:**
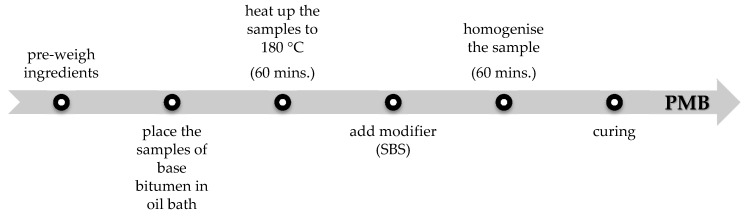
Preparation of lab samples of polymer-modified bitumens.

**Figure 3 materials-13-05237-f003:**
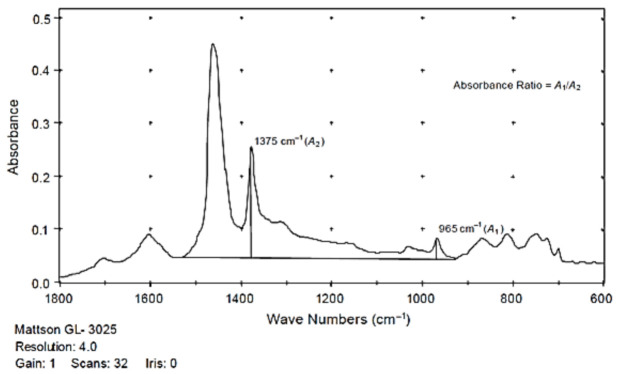
Example determination of A1 and A2 values in the transmission mode of analysis, according to AASHTO T302–15 [32].

**Figure 4 materials-13-05237-f004:**
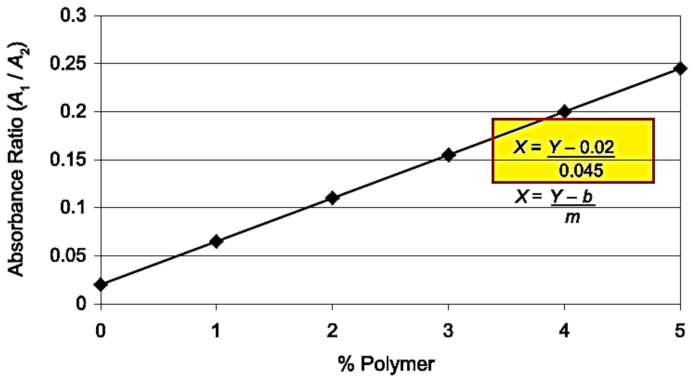
Calibration curve for determination of the percentage concentration of polymer in the transmission mode of analysis, according to AASHTO T302–15 [32].

**Figure 5 materials-13-05237-f005:**
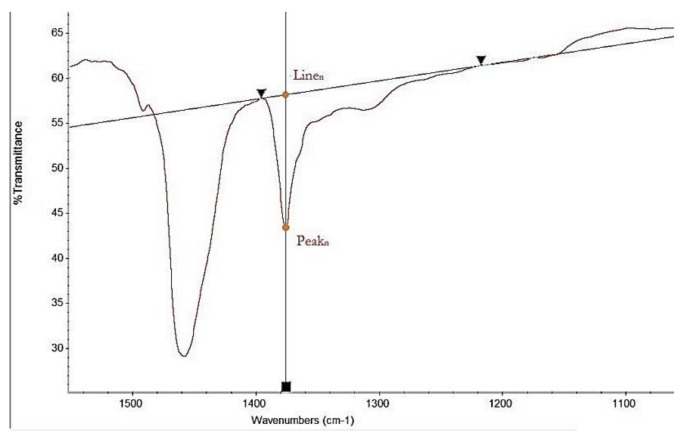
Example determination of Line_n_ and Peak_n_ percent transmission.

**Figure 6 materials-13-05237-f006:**
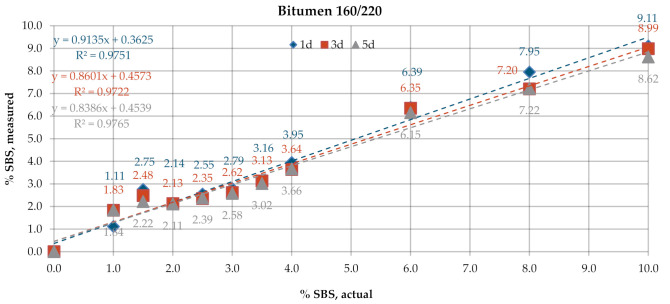
SBS contents in bitumen 160/220, obtained by measurement in transmission mode.

**Figure 7 materials-13-05237-f007:**
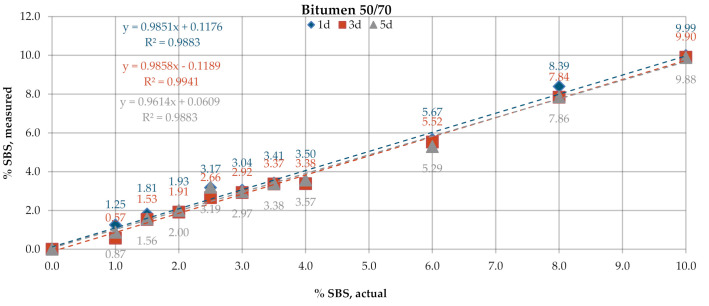
SBS contents in bitumen 50/70, transmission mode.

**Figure 8 materials-13-05237-f008:**
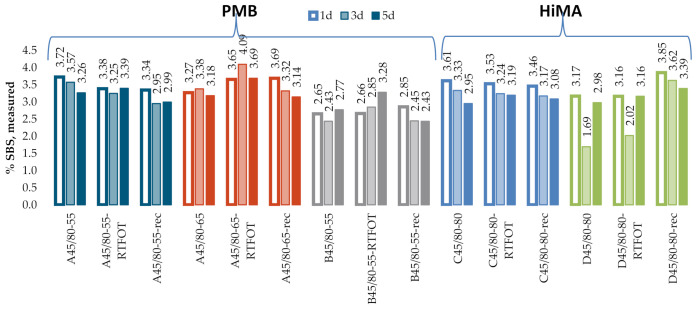
SBS contents in industrial samples, transmission mode.

**Figure 9 materials-13-05237-f009:**
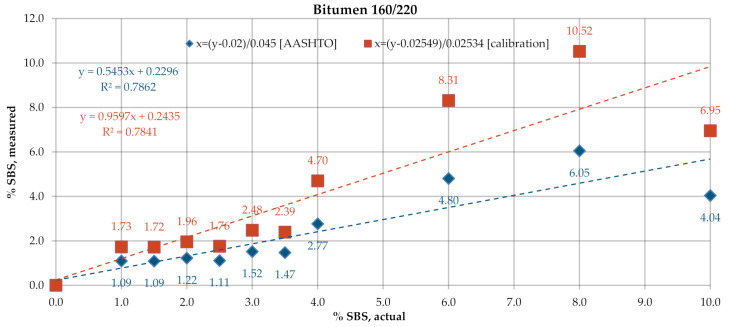
SBS contents in bitumen 160/220, ATR.

**Figure 10 materials-13-05237-f010:**
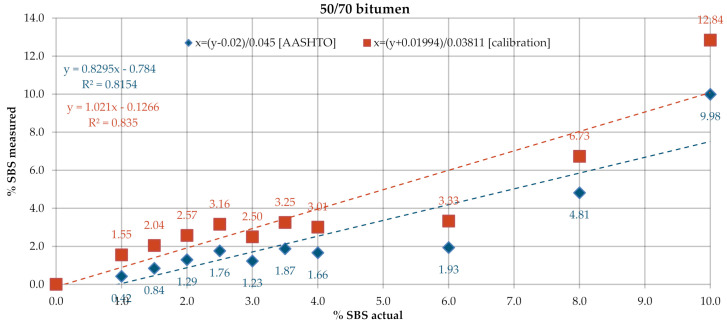
SBS contents in bitumen 50/70, ATR.

**Figure 11 materials-13-05237-f011:**
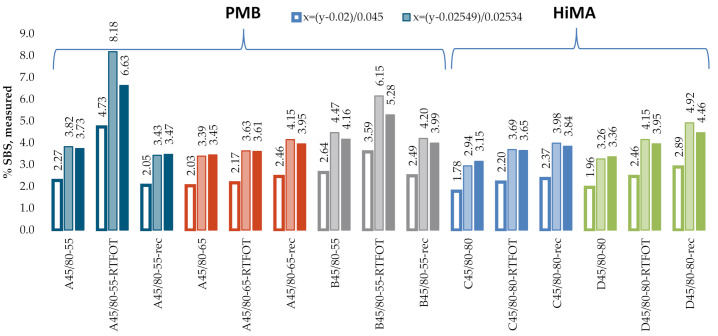
SBS contents in industrial samples, ATR (attenuated total reflectance).

**Figure 12 materials-13-05237-f012:**
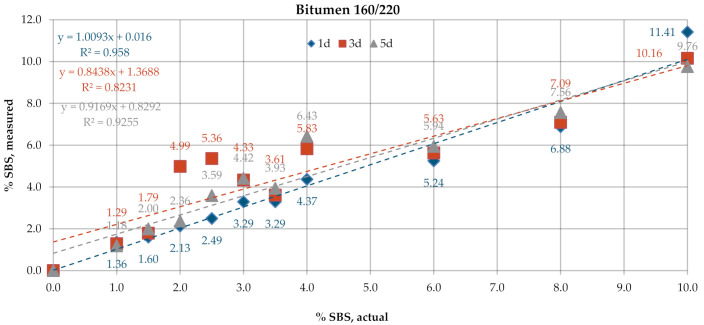
SBS contents in the samples of bitumen 160/220.

**Figure 13 materials-13-05237-f013:**
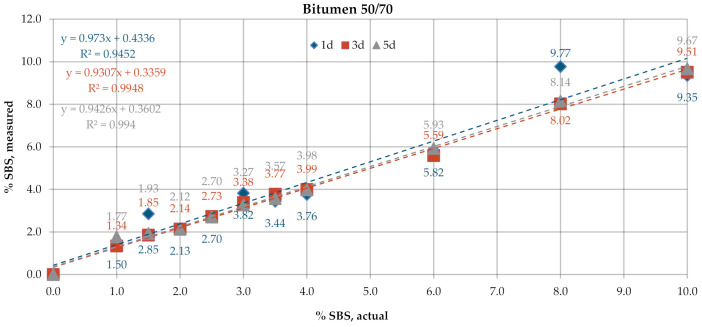
SBS contents in the samples of bitumen 50/70.

**Figure 14 materials-13-05237-f014:**
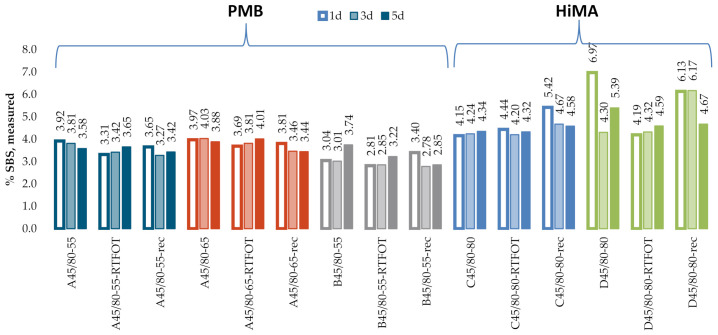
SBS contents measured on industrial samples.

**Figure 15 materials-13-05237-f015:**
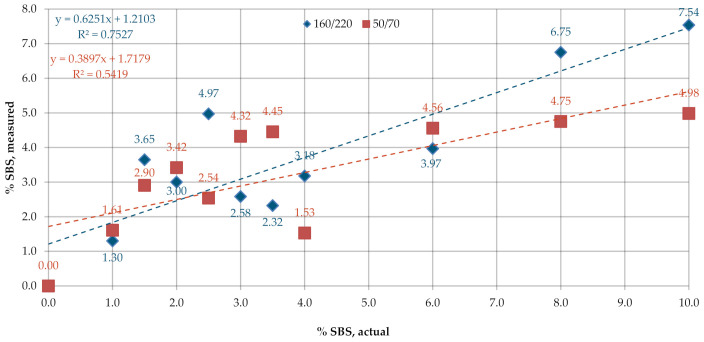
SBS contents in the samples of bitumens 160/220 and 50/70.

**Figure 16 materials-13-05237-f016:**
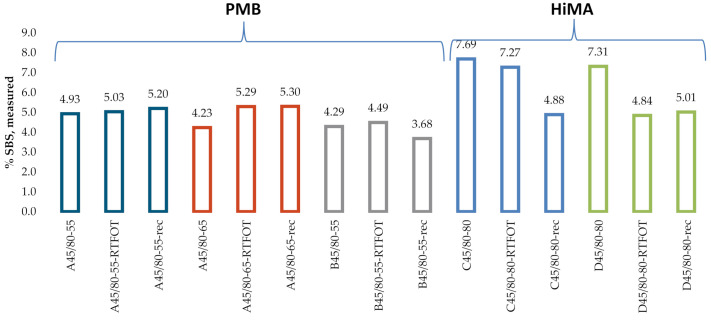
SBS contents measured on industrial samples.

**Table 1 materials-13-05237-t001:** Properties of the base bitumens.

Type of Bitumen	Softening Point (EN 1427)	Penetration at 25 °C (EN 1426)	Fraass Breaking Point (EN 12593)
-	°C	0.1 mm	°C
160/220	41.0	175	−16.0
50/70	49.6	64	−10.0

**Table 2 materials-13-05237-t002:** Properties of styrene-butadiene-styrene copolymer KRATON^TM^ D1101 A [35].

Polystyrene Content	Viscosity	Bulk Density	Melting Flow Rate (200 °C/5 kg)
%	Pa·s	kg/dm^3^	g/10 min
30 to 32	3 to 5	0.4	<1
**Tensile Strength**	**300% Modulus**	**Elongation at Break**	**Hardness, Shore A** **(30 s)**
MPa	MPa	%	HShA (30 s)
33	2.9	880	72

**Table 3 materials-13-05237-t003:** List of samples prepared for the tests and their symbols.

Bitumen Type	Symbol	Bitumen Type	Symbol
Lab samples		Industrial samples	
Bitumen 160/220	160/220	PMB 45/80-55 from Manufacturer A	A45/80-55
Bitumen 160/220 modified with 1% of SBS by weigh	160/220 + 1% SBS	PMB 45/80-55 from Manufacturer A, after simulation of long-term ageing (RTFOT)	A45/80-55-RTFOT
Bitumen 160/220 modified with 1.5% of SBS by weight	160/220 + 1.5% SBS	PMB 45/80-55 from Manufacturer A, recovered from HMA	A45/80-55-rec
Bitumen 160/220 modified with 2% of SBS by weight	160/220 + 2% SBS	PMB 45/80-65 from Manufacturer A	A45/80-65
Bitumen 160/220 modified with 2.5% of SBS by weight	160/220 + 2.5% SBS	PMB 45/80-65 from Manufacturer A, after simulation of long-term ageing (RTFOT)	A45/80-65-RTFOT
Bitumen 160/220 modified with 3% of SBS by weight	160/220 + 3% SBS	PMB 45/80-65 from Manufacturer A, recovered from HMA	A45/80-65-rec
Bitumen 160/220 modified with 3.5% of SBS by weight	160/220 + 3.5% SBS	PMB 45/80-55 from Manufacturer B	B45/80-55
Bitumen 160/220 modified with 4% of SBS by weight	160/220 + 4% SBS	PMB 45/80-55 from Manufacturer B, after simulation of long-term ageing (RTFOT)	B45/80-55-RTFOT
Bitumen 160/220 modified with 6% of SBS by weight	160/220 + 6% SBS	PMB 45/80-55 from Manufacturer B, recovered from HMA	B45/80-55-rec
Bitumen 160/220 modified with 8% of SBS by weight	160/220 + 8% SBS	HiMA 45/80-80 from Manufacturer C	C45/80-80
Bitumen 160/220 modified with 10% of SBS by weight	160/220 + 10% SBS	HiMA 45/80-80 from Manufacturer C, after simulation of long-term ageing (RTFOT)	C45/80-80-RTFOT
Bitumen 50/70	50/70	HiMA 45/80-80 from Manufacturer C, recovered from HMA	C45/80-80-rec
Bitumen 50/70 modified with 1% of SBS by weight	50/70 + 1% SBS	HiMA 45/80-80 from Manufacturer D	D45/80-80
Bitumen 50/70 modified with 1.5% of SBS by weight	50/70 + 1.5% SBS	HiMA 45/80-80 from Manufacturer D, after simulation of long-term ageing (RTFOT)	D45/80-80-RTFOT
Bitumen 50/70 modified with 2% of SBS by weight	50/70 + 2% SBS	HiMA 45/80-80 from Manufacturer D, recovered from HMA	D45/80-80-rec
Bitumen 50/70 modified with 2.5% of SBS by weight	50/70 + 2.5% SBS		
Bitumen 50/70 modified with 3% of SBS by weight	50/70 + 3% SBS		
Bitumen 50/70 modified with 3.5% of SBS by weight	50/70 + 3.5% SBS		
Bitumen 50/70 modified with 4% of SBS by weight	50/70 + 4% SBS		
Bitumen 50/70 modified with 6% of SBS by weight	50/70 + 6% SBS		
Bitumen 50/70 modified with 8% of SBS by weight	50/70 + 8% SBS		
Bitumen 50/70 modified with 10% of SBS by weight	50/70 + 10% SBS		

**Table 4 materials-13-05237-t004:** Set-up of ATR-FTIR spectrophotometer [33].

Parameter	Requirements	Parameter	Requirements
Detector	DTGS KBr	Beam splitter	KBr
Source	IR-Turbo	Accessory	Smart orbit
Window	Diamond	Gain setting	8.0
Aperture	100	Velocity	0.6329
Scan range	1100–625 cm^−1^	Number of scans	32

**Table 5 materials-13-05237-t005:** Comparative analysis of the test methods.

Test Method ^1^	Type of Modifier	Solvent	Peaks					Type of Analysis ^2^	Mean Error (Lab Sample)
			1375	965	911	808	690		
T302–15—T	SBS, SBR, SB	THF	+	+				h	0.36%
T302–15—A	SBS, SBR, SB	–	+	+				a	1.14%
T521—T	SBS, SBR	Toluene	+	+			+	h	0.66%
Q350—A	SBS	CS_2_		+	+	+	+	a	1.60%

^1^ T—transmission mode; A—attenuated total reflectance (ATR); ^2^ h—peak height; a—peak area.

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
