# Peer review of "Evaluation of Laboratory Methods of Determination of SBS Content in Polymer-Modified Bitumens"

_materials, 2020, doi:10.3390/ma13225237_

Round 1
Reviewer 1 Report
As our experience, for a given bitumen, the AFT can dictect the SBS polymer content based on their test results. But as the petrolium come from different country or different source, the pick value may be move so it is needed to check for a given bitumen and need to recheck for a different bitumen. How to use this method to keep the bitumen quality.
Author Response
Dear Reviewer,
Thank you kindly for your very helpful review. We have used the IR method because it is very widely used and easy to perform. Unfortunately we are not familiar with the AFT technique but we will be very grateful for any references. The test results presented in this paper are just a part of our research project, so any references pertaining to determining SBS content in PMBs will be very helpful. In the next step of our project we will conduct tests with bitumens obtained from different sources and containing different types of SBS to further evaluate these test methods.
The effect of ageing of bitumen can also be evaluated through IR spectroscopy [lines 124-127]. Taking this into consideration, in our research we have tested fresh bitumen obtained directly from manufacturer, binder subjected to simulated long-term ageing (RTFOT) and binder recovered from HMA. Test results show that the process of ageing of bitumen has no impact on the analyzed test methods and can be used in industry.
Please find attached the manuscript after revisions.
Kind regards,
Maria Ratajczak & Artur Wilmański

Reviewer 2 Report
This manuscript evaluates different experiment methods to quantify the SBS content in polymer modified bitumen samples, based on IR techniques.
This is a nice paper, well written, with an interesting and quite extensive experimental part. Experimental data give a reasonable interpretation of the phenomena involved in the quantification of SBS content.
However, there is main item that need to be clarified. Only one type of SBS is used for laboratory samples (KRATONTM D1101) with a linear structure and having 30-32% styrene. In my opinion, unless other polymers are evaluated, the results are only valid for SBSs similar to KRATONTM D1101. I suggest to include some examples of other SBSs with different structure (e.g radial), other styrene content, SBR vs SBS etc.
Other minor points to be clarified
- The effect of ageing needs to be discussed as authors performed an extensive experimental work including RTFOT samples.
- According to my experience, solvent evaporation methods may be altered for the surface tension effect of bitumen fractions since sometimes an O-ring of different visual composition is formed. What is the authors opinion on this issue?
- PAg 7 lines 199-201, the equations presented must be better explained.
In conclusion, I would like to recommend this manuscript for publishing after minor revision.
Author Response
Dear Reviewer,
Thank you kindly for your very helpful review. The test results presented in this paper are just a part of our research project. We have started our research with KRATONTM D1101, as it is the most widely used polymer in the Polish construction materials market. In the next step of our project we will carry out tests with bitumens obtained from different sources and with different types of SBS to evaluate the analyzed test methods. [lines 106-109] Unfortunately, at the moment we don’t have any test results for different types of SBS.
The effect of ageing of bitumen can affect the bitumen spectra and it can be evaluated through IR spectroscopy [lines 124-127].Taking this into consideration, in our research we have tested fresh bitumen obtained directly from manufacturer, binder subjected to simulated long-term ageing (RTFOT) and binder recovered from HMA. Test results show that the process of ageing of bitumen has no impact on these test methods and they can be used in industry.
The aim of this paper was to evaluate the presented test methods and all the tests were made according to the procedures described in these methods. In the AASHTO T 302-15 standard we can find an explanation of this issue: “Use of the ratio of the specified peak for the polymer to the peak for the asphalt negates any deviations due to film thickness.” In our research different film thickness were also tested and we haven’t observed any significant influence of the film thickness on the results. The surface tension effect of bitumen fractions might change the composition of bitumen and its spectra. However, if we compare the bitumen spectra made with the ATR technique and these made with transmission mode there are no differences between the peak positions but only in their heights.
Please find attached the manuscript after revisions.
Kind regards,
Maria Ratajczak & Artur Wilmański

Reviewer 3 Report
The paper is well written. The results of research are very useful and practical for industry and academia. However, there are some comments as follow:
1- The authors do not propose which methodology is the best under the identical condition. I recommend to add some comments.
2- If the source of binder changes, it may have any effect on the selection of the best method or not?
3- I recommend to remove Table 5 and pertinent discussions from the conclusion and add into the section of discussion. Just core results of research should be written in the conclusion.
Author Response
Dear Reviewer,
Thank you kindly for your very helpful review.
The T302–15 test method with transmission mode produces very satisfactory test results for laboratory samples, but with industrial samples we can observe no differences between PMB and HiMA, which is questionable. The Q350 and T302–15 ATR test methods are characterized by a high mean error value. Based on our test results we recommend the T521 test method for use. The mean error is satisfactory and the test results for industrial samples are reliable. [lines 368-372]
The test results presented in this paper are just a part of our research project. In the next step of our project we will carry out tests with bitumens from different sources and different type of SBS to evaluate these test methods. [lines 106-109]
Please find attached the manuscript after revisions.
Kind regards,
Maria Ratajczak & Artur Wilmański

This manuscript is a resubmission of an earlier submission. The following is a list of the peer review reports and author responses from that submission.